# Crucial Role of Extracellular Vesicles in Bronchial Asthma

**DOI:** 10.3390/ijms20102589

**Published:** 2019-05-27

**Authors:** Tatsuya Nagano, Masahiro Katsurada, Ryota Dokuni, Daisuke Hazama, Tatsunori Kiriu, Kanoko Umezawa, Kazuyuki Kobayashi, Yoshihiro Nishimura

**Affiliations:** Division of Respiratory Medicine, Department of Internal Medicine, Kobe University Graduate School of Medicine, 7-5-1 Kusunoki-cho, Chuo-ku, Kobe 650-0017, Japan; mskatsu@med.kobe-u.ac.jp (M.K.); rdokuni@med.kobe-u.ac.jp (R.D.); dhazama@med.kobe-u.ac.jp (D.H.); tkiriu@med.kobe-u.ac.jp (T.K.); kanoko03@med.kobe-u.ac.jp (K.U.); kkoba@med.kobe-u.ac.jp (K.K.); nishiy@med.kobe-u.ac.jp (Y.N.)

**Keywords:** extracellular vesicles, exosomes, microparticles, apoptotic bodies

## Abstract

Extracellular vesicles (EVs) are circulating vesicles secreted by various cell types. EVs are classified into three groups according to size, structural components, and generation process of vesicles: exosomes, microvesicles, and apoptotic bodies. Recently, EVs have been considered to be crucial for cell-to-cell communications and homeostasis because they contain intracellular proteins and nucleic acids. Epithelial cells from mice suffering from bronchial asthma (BA) secrete more EVs and suppress inflammation-induced EV production. Moreover, microarray analyses of bronchoalveolar lavage fluid have revealed that several microRNAs are useful novel biomarkers of BA. Mesenchymal stromal cell-derived EVs are possible candidates of novel BA therapy. In this review, we highlight the biologic roles of EVs in BA and review novel EV-targeted therapy to help understanding by clinicians and biologists.

## 1. Introduction

Extracellular vesicles (EVs) were first described as “platelet products” in human plasma in 1967 [1]. EVs were discovered to be associated with intercellular transport of trophic substances or nutrients in 1980 [2]. EVs have been called different names in different reports: “exosomes”, “microvesicles”, “ectosomes”, “oncosomes”, and “cytoplasts” [3]. György et al. classified EVs into exosomes, microvesicles, and apoptotic bodies according to size, structural components, and generation process of vesicles [4]. 

The basic features of exosomes, microvesicles, and apoptotic bodies have been covered recently in numerous review articles [4]. Briefly, exosomes are vesicles surrounded by a phospholipid bilayer and are approximately 50–100 nm in diameter. Exosomes are generated by exocytosis of multivesicular bodies from immune cells and tumor cells [5]. Exosomes have essential roles in normal cellular function and homeostatic regulation of the host tissues and organs [6,7,8,9,10,11,12]. Exosomes can be isolated from bodily fluids including blood [13], urine [14], semen [6], breast milk [15], and bronchoalveolar lavage fluid (BALF) [16]. Microvesicles are also surrounded by a phospholipid bilayer and are 100–1000 nm in diameter [17]. Microvesicles are generated by regulated release due to budding/blebbing of the plasma membranes of platelets, red blood cells, and endothelial cells [18]. Apoptotic bodies are 1–5 μm in diameter and released as blebs of cells undergoing apoptosis [19]. Apoptotic bodies are also surrounded by a phospholipid bilayer. The International Society for Extracellular Vesicles (http://www.isev.org) also recommends use of the term “EVs”. Therefore, in this review, we use “EVs” to avoid confusion or ambiguity. 

EVs contain intracellular proteins (receptors, transcription factors, enzymes, extracellular-matrix proteins) and nucleic acids including DNA, messenger RNA (mRNA), microRNA (miRNA), and noncoding RNA. They serve as mediators of intercellular communication and have a crucial role in cell-to-cell communications and homeostasis [20,21]. EVs have been extensively studied and reported to be associated with a wide range of inflammatory diseases [22,23,24] and respiratory diseases [25,26,27,28]. Hence, EVs have received considerable attention as novel biomarkers and therapeutic targets. Indeed, it has been reported that intranasal administration of EVs in BALF collected from experimental mice suffering from pollinosis induces immune tolerance and suppresses allergic reactions [29]. In an experimental model of bronchial asthma (BA) in mice, EVs were secreted from epithelial cells and suppressed production of inflammation-induced EVs [30]. This result suggested that EVs are useful for BA treatment. Moreover, by using microarray analyses of BALF, 16 miRNAs containing let-7 and miRNA-200 seem to be useful as novel biomarkers of BA [31]. Although the biological functions of EVs and their association with disease are not fully elucidated, EVs are thought to have a role in cellular homeostasis through the autophagy pathways in a complex intercellular and systemic messaging system [11,32]. In this review, we summarize recent findings of EV-mediated BA pathogenesis and discuss the potential clinical usefulness of EVs as biomarkers and therapeutic agents for BA.

## 2. EVs in BA Pathogenesis

### 2.1. EV-Secreting Cells in the Respiratory System

EV-secreting cells are summarized in Table 1. In the respiratory system, EVs are secreted from bronchial epithelial cells (BECs), alveolar macrophages, vascular endothelial cells, and fibroblasts [30]. In particular, BECs are considered to be major sources of pulmonary EVs [30]. Environmental stimuli such as cigarette smoke induce BEC injury and induce release of proinflammatory cytokines and chemokines from injured BECs [33]. Human epithelial cell lines, BEAS-2B, secrete full-length CCN1, the first member of the CCN protein family, and facilitate the interleukin (IL)-8 and vascular endothelial growth factor (VEGF) release in response to cigarette smoke extract (CSE) [34]. In addition, BECs secrete various sizes of EVs which contain different membrane-tethered mucins [35,36]. BEC-derived mucins including mucin (MUC)-1, MUC-4, and MUC-16 are present at the surface of BEC-derived EVs and have a protective role including neutralization of human influenza A virus infection by mucin-contained alpha 2, 6-sialic acids in Madin–Darby canine kidney (MDCK) cells [36]. A recent study showed that 80% of EVs in BALF from allergen-untreated mice were secreted from airway-lining epithelium by using cell-type-specific membrane tagging and single vesicle flow [37]. After induction of allergic airway inflammation by using ovalbumin (OVA), the number of immune cell-derived EVs including miR-223 and miR-142a in BALF from allergen-treated mice increased by more than twofold [37].

Macrophages are also major sources of pulmonary EVs [30]. Macrophages secrete EVs, which induce macrophage differentiation through miRNA-223 transfer [38]. Alveolar macrophages infected with mycobacteria secrete EVs which contain pathogen-derived proinflammatory molecules and secrete heat-shock proteins (HSP)70, which activates the nuclear factor-κB (NF-κB) pathway by stimulating toll-like receptors (TLRs) [44], leading to the secretion of proinflammatory cytokines [45,46]. Moreover, alveolar macrophages secrete EVs that contain suppressors of cytokine signaling (SOCS)-1 and SOCS-3, resulting in inhibition of Janus kinase and signal transducers and activators of transcription signaling [47]. Macrophages secrete the procoagulant EVs in response to CSE [48]. CSE-induced macrophage-derived EVs upregulate the synthesis of proinflammatory mediators including IL-8, intercellular adhesion molecule-1 (ICAM-1), and monocyte chemotactic protein-1 (MCP-1) by lung epithelial cells [49]. CSE-induced macrophage-derived EVs have gelatinolytic and collagenolytic activities that are associated with a single transmembrane protease of the matrix metalloproteinase (MMP) superfamily [50].

Endothelial cells also secrete various types of EVs, including endothelial microparticles, which contain platelet endothelial cell adhesion molecule-1, vascular endothelial cadherin, and E-selectin, and have roles in coagulation, inflammation, endothelial function, and angiogenesis [33]. Activated human lung fibroblasts secrete EVs containing several prostaglandins (PGs), including high levels of the antifibrotic PGE2, and have roles in preventing myofibroblast differentiation and maintaining homeostasis [51]. 

Besides the cells mentioned above, it has been reported that B cells secrete EVs and have roles as immune-system activators [40]. Also, dendritic cells (DCs) secrete EVs and have roles in modulating immune reactions by activating T and B cells [41,42]. These findings suggest that EVs could regulate airway inflammation and allergic reactions through intercellular communications.

Mesenchymal stromal cells (MSCs) are recruited to inflammation sites [52], stimulate endogenous repair of injured tissues, and modulate immune response [53]. MSCs secrete EVs which are enriched in mRNAs [54]. For example, these EVs express mRNA for transcription factors including *MDFIC*, *POU3F1*, and *NRIP1*, and genes involved in angiogenesis including *HGF*, *HES1*, and *TCF4* and genes involved in adipogenesis including *CEBPA* and *KLF7* [55].

### 2.2. Allergen-Induced Airway Inflammation

Allergic inflammation is initiated with DC activation by allergens [56]. Activated DCs mature and present peptides derived from the processed allergen by major histocompatibility complex (MHC) class-II molecules to naïve T cells. DCs secrete the chemokines CC-chemokine ligand 17 (CCL17) and CCL22, which act on CC-chemokine receptor 4 to attract T helper 2 (T_H_2) cells [57]. T_H_2 cells produce IL-4 and IL-13, which stimulate plasma B cells to secrete allergen-specific immunoglobulin E (IgE). Secreted IgE binds to the high-affinity receptor for IgE on tissue-resident mast cells in a process known as “sensitization”. Thereafter, mast cells release three classes of biologically active product when they are re-exposed to the allergen: cytoplasmic granules (e.g., histamine) in a process known as “degranulation”, lipid-derived mediators (e.g., PGs, leukotrienes), and cytokines, chemokines, and growth factors. Finally, these events cause vasodilation, increased vascular permeability, bronchoconstriction, and mucus secretion from airways [56]. Lipid-derived mediators drive airway inflammation, promote immune cell infiltration, and induce mucus hyperplasia. Indeed, leukotrienes promote mucus secretion, smooth-muscle contraction, and airway inflammation [58]. Ceramides and PGs also have a role in inflammation in the airways of BA patients [59,60,61,62]. T_H_2 cells produce IL-5, which has a crucial role in the differentiation, activation, and survival of eosinophils [63]. Eosinophils are associated with frequent exacerbations and fixed airflow limitation [64]. BA is orchestrated by IL-4-induced IgE-production and IL-5-induced expansion of eosinophils, which release proinflammatory and bronchoconstricting granular content [65]. When allergen exposure is repeated, innate immune cells including eosinophils, basophils, neutrophils, and monocytes, and adaptive immune cells including T and B cells take up residence in the inflammatory site. Then, complex interactions between innate immune cells and adaptive immune cells and lung structural cell are initiated, resulting in bronchoconstriction and severe narrowing of airway lumen. In some cases, T_H_17 also contributes to the recruitment of neutrophils to the inflammation site. 

A recent study clarified the important role of regulatory T cells (T_reg_) in BA development. Indeed, induced T_reg_ cells suppress production of group-2 innate lymphoid cell-driven IL-5 and IL-13 [66]. Furthermore, recent studies have shown that IL-9, which is secreted by T_H_9 cells, is critically involved in the immune-pathogenesis of inflammatory diseases including BA and in guarding immune tolerance [67].

### 2.3. Functions of EVs in BA Pathogenesis

#### 2.3.1. EVs From Hematopoietic Cells

EVs have a role in development, recruitment, activation, and suppression of the immune system [9,17,40,68,69,70,71,72,73,74]. Indeed, peripheral DCs secrete EVs, which are endocytosed and re-presented on the cell surface or transfer MHC/peptide complexes to recipient DCs after antigen uptake and processing [17,75,76,77,78]. These DC-derived EVs can activate T cells with assistance from DCs and B cells [79,80]. On the other hand, B cells also secrete EVs which express MHC, bind Bet v1-derived peptides, and, subsequently, induce dose-dependent T cell proliferation [81]. The surface of B-cell-derived EVs contains clusters of differentiation (CD)40, CD80, and CD86, which have costimulatory capacity as well as integrins (α1 and α2) and enable B cells to exert important effects over T cell response [72,82,83]. Follicular DCs that lack the expression of MHC class II molecules received peptide-bound MHC class II molecules from EVs secreted by B cells [84,85]. Adhesion of EVs to the surface of follicular DCs is through the oligomerization and binding of tetraspanins between the EVs and follicular DCs [86]. In particular, ICAM-1, also known as CD54, facilitates this adhesion of EVs [16,87,88]. CD8^+^ DCs capture MHC-peptide complexes from EVs by the ligand for CD54, lymphocyte function-associated antigen-1 (LFA-1) [77]. Moreover, CD54/LFA-1 interactions on DCs are associated with internalization of EVs in immune cells [89]. Another study revealed that LFA-1 has a role in recruitment of EVs to T cells and their subsequent activation [90]. These findings suggest that EVs can modulate immune memory through expanding the repertoire of antigens [25]. 

EV production of monocyte-derived macrophages is modulated by transforming growth factor (TGF)-β, IL-1β, and interferon (IFN)-γ from airway smooth muscle. These cytokines affect the rate of exosome generation and delivery by peripheral blood monocytes or alveolar macrophages [30,38,91]. DCs and macrophages in humans secrete EVs which contain enzymes for leukotriene biosynthesis and which induce granulocyte migration [91]. This observation suggests that EVs have a proinflammatory role. Macrophages and DCs also secrete EVs which contain proinflammatory lipid mediators such as leukotrienes and promote migration of granulocyte [91]. Platelets also secrete EVs which transfer the leukotriene precursor arachidonic acid and induce leukotriene production in recipient platelets and endothelial cells [92]. Platelet-derived EVs promote adhesion of neutrophils to endothelial cells through CD62P and CXC chemokines [93]. Activated platelets adhere to intravascular neutrophils through P-selectin/P-selectin glycoprotein ligand-1 (PSGL-1)-mediated binding and allow platelets glycoprotein Ibα (GPIbα)-induced generation of neutrophil-derived EVs, which in turn synthesize thromboxane A2 [94]. Finally, platelet-derived thromboxane A2 elicits a full neutrophil response by inducing the endothelial expression of ICAM-1 and promotes neutrophil extravasation.

Eosinophils purified from peripheral blood can secrete EVs by IFN-γ stimulation [95]. These EVs contain eosinophil-derived enzymes including eosinophil peroxidase (EPO), major basic protein (MBP), and eosinophil cationic protein (ECP) and have a role in BA.EVs from eosinophils of patients with BA act in an autocrine manner and can be distinguished from EVs from eosinophils of healthy subjects. Moreover, eosinophil-derived EVs increase reactive oxygen species (ROS) and nitric oxide (NO) production augmented chemotaxis of eosinophil and adhesion by upregulating ICAM-1 and integrin-α2 [96]. EVs from eosinophils of patients with BA can modify the behavior of small airway epithelial cells, increase their apoptosis, reduce their wound healing capacity, and enhance the expression of *CCL26*, *TNF*, and *POSTN* in epithelial cells and *CCR3* and *VEGFA* in primary bronchial smooth muscle cells [97]. Neutrophil can also secrete EVs by lipopolysaccharides (LPS) stimulation [98]. These EVs are internalized by airway smooth muscle cells and their proliferative properties are altered. Furthermore, EVs containing leukotriene B4 act in an autocrine manner to sensitize neutrophils towards the primary chemoattractant, and in a paracrine manner to mediate the recruitment of neighboring neutrophils in trans [99].

#### 2.3.2. EVs from Lung Structural Cells

EVs from pulmonary epithelial cells metabolize myeloid cell-derived leukotriene C4 to leukotriene D4 [100]. Furthermore, ceramides and sphingolipids are contained in EVs and have a role in inflammation [101,102]. In murine model of macrophage-mediated hepatic inflammation, ceramides of EVs act as chemoattractants of macrophages [103]. Tumor necrosis factor (TNF)-α and IFN-γ induce the oligodendroglioma cell lines release of ceramide-enriched EVs as a mediator of cell death signaling [104]. IL-13 also induces EV secretion from epithelial cells, increasing chemotaxis and proliferation of macrophages [30]. Mechanical stress (e.g., bronchoconstriction) leads BECs to produce EVs, which have a role in promotion of subepithelial fibrosis and angiogenesis [105]. BEC-derived EVs also drive proliferation of monocytes and enhance chemotaxis [30]. EVs in BALF from BA patients express more clusters of differentiation-36, which has a role in bacterial infection-induced BA exacerbations through bacterial recognition compared with EVs in BALF from healthy controls [106]. T_H_2 cytokines can stimulate BEC production of EVs and induce monocyte proliferation. Moreover, monocytes which are treated with BEC-derived EVs enhance their migration in the presence of MCP-1 [25]. In addition to host cell-derived EVs, microbial EVs have a role in immune activation and hypersensitivity [107,108,109].

#### 2.3.3. Role of miRNA in EVs

EVs can package miRNAs which have important roles in BA. miRNAs are small, noncoding RNAs, 20 to 23 nucleotides in length, packed within secreted EVs, and which regulate gene expression by destabilizing and degrading mRNAs [110,111,112,113,114]. A distinct set of extracellular miRNAs in induced sputum and BALF has also been identified as being associated with the respiratory tract [31,115,116]. In a rat study, serum EVs were identified with 16 different proinflammatory miRNAs in response to zinc oxide nanoparticles [117]. Additionally, miR-155 has a role in the development of infiltration of inflammatory cells into the lungs and in airway remodeling [118]. miR-155 expression is increased significantly in the lungs after allergen exposure, and miR-155-deficient mice have diminished eosinophilic inflammation in the lungs, suggesting that miR-155 is essential for T_H_2 cell-mediated inflammation [119]. miR-126 has a role in the effector function of T_H_2 cells, and antagonism of miR-126 suppresses allergic airway inflammation [120,121]. miR-27 and miR-24 are also important for T_H_2 response [122]. On the other hand, mir-21 has a role in metabolic regulation of pathogenic T_H_17 [123]. miR-221 has a role in mast-cell degranulation and cytokine production [124] and miR-221 blockade suppresses airway inflammation [125]. Furthermore, downregulation of miR-133a expression causes upregulation of RhoA expression, resulting in augmentation of airway contraction and hyperresponsiveness [126]. These findings suggest that EVs containing miRNA can modulate gene programming and promote inflammation in an antigen-independent manner.

#### 2.3.4. Novel Function of EVs

Recent studies revealed that EVs have a role in transferring mitochondria, besides transferring bioactive materials [127,128,129]. Myeloid-derived regulatory cells transfer EVs containing mitochondria to peripheral T cells and regulate T cell responses in asthma [129]. MSC-derived EVs which contain mitochondria promote M2 polarization of macrophages and oxidative phosphorylation [128]. This novel finding suggests that EVs have a role in novel cell–cell communication involving EV transfer of mitochondria and the bioenergetics regulation of target cells.

## 3. EVs as Potential Biomarkers of BA

EVs in BALF from BA patients contain miRNAs and could be biomarkers of BA [31]. EV concentrations in BALF from BA patients increase and correlate with blood eosinophilia and serum IgE levels [88]. Several studies have revealed that the numbers of human leukocyte antigen D-related (HLA-DR)^+^ and ICAM-1^+^ EVs also increase in BALF from BA patients [16,87,88,130]. HLA-DR is a class-II antigen-presentation molecule on antigen-presenting cells [131]. ICAM-1 is an adhesion molecule and is expressed on the activated bronchial epithelial cells of BA patients [132]. Lipidomics has shown that levels of phosphatidylglycerol, ceramide-phosphates, and ceramides decrease in EVs in the BALF from BA patients [88]. Differential centrifugation with or without a gradient is a well-established purification method for EVs [133]. Commercial EV purification kits are also available [134]. Tetraspanins including CD63, CD9, CD81, CD82, HSP70, HSP90, tumor susceptibility gene 101 (TSG101), ALG-2 interacting protein X (ALIX), actin, and glyceraldehyde-3-phosphate dehydrogenase (GAPDH) are used as cell surface markers to identify EVs [135,136,137]. Generally, tetraspanins, integrins, cell-adhesion molecules, proteoglycans, and lectins are associated with cell–cell interactions and triggering signaling transduction [138]. Endosomal sorting complexes required for transport (ESCRT) including TSG101, CD81, and ALIX are used as endosomal markers [110,139]. Surprisingly, it has been reported that EVs in exhaled breath condensates (EBCs) from BA patients can be noninvasive biomarkers of BA [140,141]. However, miRNAs in serum are more suitable biomarkers than those in BALF or EBCs in clinical settings [142]. 

## 4. EVs as Potential Therapeutic Targets for BA

EVs in the BALF of patients with BA increased IL-4 and leukotriene C4 in BECs [87]. Administration of BALF-derived EVs from experimental mice with BA inhibits IgE responses, T_H_2 cytokine production, and airway inflammation [29]. EV secretion has been reported to be prevented by GW4869, which inhibits neutral sphingomyelinase-2 (a key regulatory enzyme that generates ceramide from sphingomyelin) [143,144,145,146,147,148,149,150,151,152]. In the murine house dust mite (HDM)-induced asthma model, EVs in BALF from HDM-exposed mice are 8.9-fold higher than those from sham-control mice [153]. HDM induced significant changes in the expression of 139 miRNAs in EVs and upregulated 31 genes including IL-13 and IL-5Ra that are putative targets of the miR-346 and miR-574-5p, respectively. GW4869 administration reduces EV secretion and ameliorates BA in mice [30,153]. Systemic administration of EVs secreted by human MSCs ameliorates eosinophilic and neutrophilic airway allergic inflammation in experimental mice [154]. EVs secreted by MSCs promote the proliferation and immune-suppression capacity of T_reg_ cells by upregulating expression of IL-10 and transforming growth factor-β1 from peripheral blood mononuclear cells [155]. Two safety studies of MSCs and their trophic factors in BA patients are in progress (NCT02192736 and NCT03137199).

EVs are useful in drug delivery, since EVs can reduce toxicity, carry both lipophilic and hydrophilic drugs to target cells, and preserve drugs’ therapeutic activity [156,157]. In order to preserve drugs’ therapeutic activity, two distinct methods of loading molecular cargo into exosome have been developed. Briefly, passive methods incubate hydrophobic agents and EVs. On the other hand, active methods use extrusion, sonication, or electroporation techniques. Mast cell- and DC-derived EVs display a specific lipid composition and are stable in circulation [158] and are protected from complement-mediated lysis by expression of CD55 and CD59 [159]. Moreover, EVs can deliver therapeutic biomolecules ranging from nucleic acids to small molecules. For example, EVs loaded with curcumin lead to an increase in the stability of curcumin in vitro and bioavailability in vivo [160]. Genetically engineered EVs carrying suicide gene mRNA and protein–cytosine deaminase fused to uracil phosphoribosyltransferase reduce schwannoma tumor growth [161]. Targeted drug delivery vesicles with low immunogenicity and toxicity were developed by electroporation technology [162]. This approach has been used to load small-molecule drugs and small interfering RNA (siRNA) [163]. However, electroporation of EVs with siRNA requires careful attention, since electroporation causes extensive siRNA aggregate formation, resulting in overestimation of the amount of siRNA actually loaded into EVs [164]. Unique trehalose pulse media is reported to minimize exosome aggregation following electroporation [165], suggesting that it may be possible to reduce aggregation of EVs and cargo RNA by optimizing electroporation conditions. 

Another strategy for loading therapeutically active cargo molecules into EVs uses hydrophobic agents, which are incubated with EVs or donor cells [166]. Interestingly, most of the cells incubated with chemotherapeutic agents or nucleic acids can package these molecules into EVs. Indeed, DCs have been used in several experimental settings as EV donor cells due to their low immunogenicity profile. EVs released from indoleamine 2,3-dioxygenase (IDO)-expressing DCs have anti-inflammatory effects, and are able to reverse arthritis in a murine model of collagen-induced disease [167]. MSCs are promising sources of EVs for drug delivery in the treatment of several disorders, since they can repair tissues and have immunomodulatory properties. Indeed, intravenous delivery of MSC-derived EVs improves functional recovery and promotes neuroplasticity in young adult male rats [168]. Viral packing strategy is one of the approaches for loading nucleic acids into EVs. Nonenveloped viruses such as adeno-associated virus [169] and hepatitis A virus [170] can be incorporated into EVs during propagation.

The technologies for EV isolation are critical for future applications of EVs as drug delivery vehicles. The differential centrifugations followed by ultracentrifugation are commonly used to remove cells and large cell debris and precipitate EVs [133,171]. Another EV isolation technique is based on precipitation by polymers [172,173,174] or size exclusion chromatography (SEC) [175,176]. SEC has proven to be beneficial in the elimination of contaminants such as proteins and lipoproteins [177]. Lastly, ultrafiltration involving the use of membranes with specific pore size is a suitable approach for rapid EV purification from large volume samples and allows a high degree of purification when it is combined with SEC [178,179,180].

## 5. Conclusions

In the respiratory system, EVs are secreted from all types of hematopoietic cells and lung structural cells and have a role in intercellular communication, modulating antigen presentation and immune activation, suppression, and surveillance. BA is a chronic inflammatory disease of the airways with a complex pathophysiology that involves hematopoietic cells, including DCs, T and B cells, monocytes, and eosinophils, and lung structural cells. Several studies revealed EVs in BALF and EBC from BA patients could be biomarkers of BA. EVs are potential therapeutic targets for BA and two safety studies of MSCs and their trophic factors in BA patients are in progress. Although EVs are biologically well studied and are promising biomarkers and therapeutic targets for asthma, several clinical trials are required for clinical application.

## Figures and Tables

**Table 1 ijms-20-02589-t001:** The characteristics of extracellular vesicles in the respiratory system.

Source	Function	Reference
Bronchial epithelial cells	Regulate the normal airway biology including homeostasis and innate defense	[36]
Macrophages	Maintain homeostasis and immune cell production	[38]
Endothelial cells	Activate neighboring pericytes through proinflammatory microRNAs	[39]
B cells	Activate immune system through antigen presentation	[40]
Dendritic cells	Modulate immune reactions through activating T and B cells	[41,42]
Mesenchymal stromal cells	Modulate the polarization of macrophages	[43]

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
