# Peer review of "Crucial Role of Extracellular Vesicles in Bronchial Asthma"

_ijms, 2019, doi:10.3390/ijms20102589_

Reviewer 1 Report

In this review article the authors collect the most updated knowledge on the role of Extracellular Vesicles (EVs) in Bronchial Asthma. The review is well organized with an initial part describing the different types of EVs and the source cells, a part dedicated to the use of EVs in diagnosis and the third on therapy. My general evaluation is positive; however, I would suggest some minor improvements before publication.

1) The review is missing any future perspective on the topic. For instance, the reader would be interested to know which knowledge is missing in the biology of EVs or if there are possible novel and unexplored possibilities for EVs in diagnosis and treatment of BA.

2) Line 64, “secrete” should be “secreting”

3) Line 121, “derive” should be “drive”

4) Line 228 there is a typo: “6bioenergetics”

Author Response

1) The review is missing any future perspective on the topic. For instance, the reader would be interested to know which knowledge is missing in the biology of EVs or if there are possible novel and unexplored possibilities for EVs in diagnosis and treatment of BA.

Response 1): At first, we really thank you for constructive comments that help us to improve the quality of our presentation. We hope that the changes we made are enough to make this manuscript acceptable for publication in International Journal of Molecular Sciences. We agreed your comment 1) and added the following sentences in conclusions: Although EVs are biologically well-studied and are promising biomarkers and therapeutic targets for asthma, several clinical trials are required for clinical application. The revised version was created using the track change mode in MS-Word to show the changes made.

2) Line 64, “secrete” should be “secreting”

Response 2): We thank this comment and change “secrete” into “secreting”.

3) Line 121, “derive” should be “drive”

Response 3): We thank this comment and change “derive” into “drive”.

4) Line 228 there is a typo: “6bioenergetics”

Response 4): We thank this comment and change “6bioenergetics” into “bioenergetics”.

Reviewer 2 Report

Very well written paper, I would like to add some comments:

-Can you explain if Apoptotic bodies have any phospholipid bilayer or not in line 41?

-In page 191, briefly mention which murine model of inflammation,

-In line can you explain how EVs are preserving drugs therapeutic activity?

Author Response

-Can you explain if Apoptotic bodies have any phospholipid bilayer or not in line 41?

Response: We really thank for your time despite your busyness. We thank this comment and add the following sentence in introduction: Apoptotic bodies are also surrounded by a phospholipid bilayer..

-In page 191, briefly mention which murine model of inflammation,

Response: We thank this comment and change “inflammation” into “macrophage-mediated hepatic inflammation”.

-In line can you explain how EVs are preserving drugs therapeutic activity?

Response: We thank this comment and add the following sentence: In order to preserve drugs therapeutic activity, two distinct methods of loading molecular cargo into exosome have been developed. Briefly, in passive methods incubate hydrophobic agents and EVs. On the other hand, active methods use extrusion, sonication, or electroporation techniques..